# Using A Protoplast Transformation System to Enable Functional Studies in *Mangifera indica* L.

**DOI:** 10.3390/ijms241511984

**Published:** 2023-07-26

**Authors:** Mark Owusu Adjei, Huan Zhao, Xiaoguang Tao, Li Yang, Shuyue Deng, Xiyan Li, Xinjing Mao, Shujiang Li, Jianfeng Huang, Ruixiong Luo, Aiping Gao, Jun Ma

**Affiliations:** 1College of Landscape Architecture, Sichuan Agricultural University, Chengdu 611130, China; 2College of Forestry, Sichuan Agricultural University, Chengdu 611130, China; 14087@sicau.edu.cn; 3Tropical Crop Genetic Resources Institute, Chinese Academy of Agricultural Science, Haikou 571101, China

**Keywords:** *Mangifera indica* L., protoplast isolation, transformation, polyethylene glycol mediated, transient expression

## Abstract

Mangoes (*Mangifera indica* L.) are an important kind of perennial fruit tree, but their biochemical testing method and transformation technology were insufficient and had not been rigorously explored. The protoplast technology is an excellent method for creating a rapid and effective tool for transient expression and transformation assays, particularly in plants that lack an Agrobacterium-mediated plant transformation system. This study optimized the conditions of the protoplast isolation and transformation system, which can provide a lot of help in the gene expression regulation study of mango. The most beneficial protoplast isolation conditions were 150 mg/mL of cellulase R-10 and 180 mg/mL of macerozyme R-10 in the digestion solution at pH 5.6 and 12 h of digestion time. The 0.16 M and 0.08 M mannitol in wash solution (WI) and suspension for counting (MMG), respectively, were optimal for the protoplast isolation yield. The isolated leaf protoplasts (~5.4 × 10^5^ cells/10 mL) were transfected for 30 min mediated by 40% calcium-chloride-based polyethylene glycol (PEG)-4000-CaCl_2_, from which 84.38% of the protoplasts were transformed. About 0.08 M and 0.12 M of mannitol concentration in MMG and transfection solutions, respectively, were optimal for protoplast viability. Under the florescence signal, GFP was seen in the transformed protoplasts. This showed that the target gene was successfully induced into the protoplast and that it can be transcribed and translated. Experimental results in this paper show that our high-efficiency protoplast isolation and PEG-mediated transformation protocols can provide excellent new methods for creating a rapid and effective tool for the molecular mechanism study of mangoes.

## 1. Introduction

Among the largest fruit crop families, mangoes (*Mangifera indica* L.) are grown in a wide range of habitats, including rainforests and cold regions [1]. Mangoes are widely used as food and medicine [2,3]. Mangoes are perennial fruit trees that are propagated vegetatively via grafting. Tissue culture and gene transformation techniques are limited, which delays the breeding and molecular mechanism understanding of mango. There is a need to develop tools for manipulating and analyzing molecular processes to promote research in various aspects of *Mangifera indica* L. [4]. The use of protoplasts to create tools for protein subcellular localization, protein-to-protein interactions, gene overexpression, and gene silencing has increased over the years [5]. Using protoplasts has been proven to be a fast, easy, and effective method to express a specific target gene without using the whole plant [6]. Protoplast transformation systems can be applied in studies on cell signaling pathways in response to hormones [7], immediate transcriptome responses, and regulatory networks [8]. Protoplast-based biochemical and molecular studies have been applied to carnations [9], cotton [10], citruses [11], eggplants [7], oil palms [12], pineapples [13], *Mangifera indica* L. [14], bananas [15], etc. This is because the versatility and ability of protoplast transient expression systems can be developed and applied to many non-model plants that have transformation platforms that are difficult to use [16]. Moreover, it has several advantages, including the delivery of several genes with high levels of co-transformation and a high frequency of transformation [17,18]. Isolating viable protoplasts and developing gene expression systems from tissues such as leaf, root, callus, flower, etc., are critical steps for establishing a reliable and feasible protoplast transformation method [19]. The efficiency of this method is affected by many factors, such as the composition of the digestion solution, the tissue types, the digestion time, the pH of the digestion solution, the polyethylene glycol (PEG) concentration, and the transformation time [20,21]. For example, the digestion solution containing osmotic stabilizers (mannitol), enzymes (cellulase R-10 and macerozyme R-10), or a combination of the two significantly affected the protoplast yields [22]. In this case, the cellulase R-10 degrades natural cellulose and dissolves plant cell walls [23], the macerozyme R-10 contains pectinase and hemicellulose that degrade the plant tissues to single cells [24], and the mannitol provides a suitable osmotic pressure and thereby ensures the integrity of the protoplasts [25]. Moreover, the digestion time affects the protoplast yield, and prolonging the digestion time may or may not increase the protoplast yield because an extended digestion time may result in protoplast breakage [15]. The isolated protoplast potency and viability rate are dependent on the digestion solution’s pH value. The pH of the digestion solution has a direct relationship with the cell wall permeability [26]. PEG is regarded as an effective protoplast transfection mediator where a high rate of protoplast transfection is achieved [27]. The protoplast transformation efficiency and the cell viability rate and potency are affected by the PEG concentration and transfection time [20].

The current biochemical testing methods and transformation technology systems were insufficient and had not been rigorously explored, which hindered the development of studies on gene expressions and molecular mechanisms in perennial woody plants [19]. The protoplast transient expression protocol and its broad usage for functional genomics studies of mangoes have not been fully tested [8]. Simplifying the protoplast preparation procedure and establishing a rapid gene expression study system are important and meaningful for the molecular mechanism study and breeding of mangoes. Here, an optimized leaf-based protoplast isolation and PEG-mediated transient expression protocol are established. This current study is important for molecular mechanism studies because the gene expression regulation at the cell level will give a clear view of how cellular mechanisms evolved, which is crucial for understanding the regulation of photosynthesis, metabolism, and other underlying plant growth mechanisms [28]. This investigation can be later used to analyze the function of cell developmental genes by observing the growth and development of transformed protoplasts [5]. The development of mango protoplast transient expression technology provides a versatile experimental platform to enable molecular, cellular, and functional studies of *Mangifera indica* L.

## 2. Results

### 2.1. Development of Protoplast Isolation System

Several factors affect mango protoplast isolation [19], and this study revealed a simple and direct process for isolating protoplasts from mango leaves. In this study, fresh *Mangifera indica* L. cv. *Keitt* leaves were cut into small pieces (Figure 1A) and immerged in a digestion solution to separate protoplasts (Figure 1B). The protoplasts were successfully released from green leaf tissues in a digestion solution and remained intact after digestion (Figure 1C,D).

To isolate the mesophyll protoplast from the leaf tissues, the digestion solution, which contained an osmotic stabilizer (mannitol) and two enzymes (cellulase R-10 and macerozyme R-10), was used. However, the digestion time also contributed to the survival and yield of the protoplast. During the first 12 h of digestion, the protoplast yield increased significantly to 6.5 × 10^5^ as the digestion time extended, but it decreased a lot when the digestion time was extended to 24 h (1.5 × 10^5^) (Figure 2A). It was revealed that leaf tissue digested with enzymes for 12 h gave a better yield than that digested for 6, 8, and 24 h (Figure 2A). Thus, below or above the optimal time, cells cannot maintain hyperosmotic movement, thereby resulting in fewer protoplast cells. A proper pH value for the digestion solution is important for protoplast isolation and survival. The effects of different pH values (5, 5.2, 5.4, 5.6, 5.8, and 6.0) of the digestion solution were evaluated by comparing the protoplast yield under 12 h of digestion. The results indicated that pH 5.6 had the highest protoplast yield (7.5 × 10^5^, Figure 2B). Moreover, the mannitol concentrations in the wash solution (WI) and suspension for counting (MMG) were important for the isolation and counting of protoplasts. The highest protoplast yield was obtained at 6.0 × 10^5^ when 0.16 M mannitol was added to the WI solution (Figure 2C). Moreover, the resuspension of protoplasts in 0.04 M and 0.12 M of mannitol decreased the protoplast yield from 4.0 × 10^5^ to 2.8 × 10^5^. Similarly, different mannitol concentrations (0.04, 0.08, 0.12, and 0.16 M) in the MMG buffer also apparently affected the protoplast yield. After resuspension in the MMG-0.08 M counting solution, a higher protoplast yield at 5.0 × 10^5^ as the optimal point was obtained (Figure 2D).

The protoplasts produced were strained using a 100µm strainer. The protoplast yield observed after hours of digestion showed that there was insufficient protoplast observed after 6 h of digestion, representing 2.0 × 10^5^ (Figure 3A). There was an optimal yield of 5.5 × 10^5^ and 6.5 × 10^5^ after 8 h extended to 12 h, respectively (Figure 3B,C). The protoplast yield further decreased when the digestion time was extended to 24 h at 1.5 × 10^5^ (Figure 3D).

In order to further evaluate the quality of the isolated protoplast, trypan blue staining was carried out to calculate the protoplast’s viability rate under different isolation conditions. After 6, 8, 12, and 24 h of digestion, the protoplast’s viability rate was detected. As the digestion time increased from 6 to 8 h, the viability rate increased from 38.46% to 84.62%. By increasing the digestion time to 12 h, the protoplast viability rate increased to 92.86%, which is the optimal point. However, extending the digestion time to 24 h decreased the protoplast viability rate to 33.33% (Figure 4A). When the mannitol concentration in the WI solution increased from 0.04 M to 0.08 M, the protoplast viability rates increased steadily from 57.14% to 76.92%. A maximum viability rate of 84.51% was attained at 0.16 M of mannitol in the WI solution (Figure 4B). In the same way, different concentrations of mannitol in suspension for counting (MMG) solutions also changed the rates of protoplast survival. The viability rate increased steadily from 33.33% to 71.43% after resuspension in MMG-0.04 M and MMG-0.08 M, as the optimal point was obtained at MMG-0.08 M but decreased to 50.00% at MMG-0.12 M. The protoplasts’ viability rate declined quickly again at MMG-0.16 M (Figure 4C). In total, the optimized protoplast isolation conditions in this study not only obtained the optimal protoplast yield but also the optimal protoplast viability rate. About 73.91 and 84.38 percent of the protoplasts in 0.04 M and 0.12 M mannitol-based solutions, respectively, were still alive after being transfected with the solution. But, at 0.08 M and 0.16 M of mannitol-based solutions, the viability rate decreased significantly (Figure 4D).

There was a significant difference in protoplast viability rate between 0.04 M and 0.12 M mannitol-based buffers, and 0.12 M mannitol worked better in protecting protoplasts during the transient expression procedure (Figure 4D). Therefore, 0.12 M of mannitol was chosen for the rest of the experiments. Under this condition, the yield of viable protoplasts after transfection was estimated to be approximately 5.4 × 10^5^ cells in 10 mL, representing 84.38% of the viability rate of transfected protoplasts prepared in the 0.12 M mannitol condition.

### 2.2. Development of Protoplast Transfection System

To determine the optimal conditions for mango protoplast transfection, a pC2300-GFP vector (https://www.snapgene.com/plasmids/plant_vectors/pCAMBIA2300) (accessed on 20 April 2022) was used for transformation. The transfection conditions were optimized under different rates of PEG-4000-CaCl_2_ medium (20, 30, 40, 50, 60, and 70%), and different transfection periods were optimized at various times (5, 10, 20, 30, 40, and 60 min) before microscopic observation (Figure 5A,B). The concentration of isolated protoplasts used for transformation was approximately 5.4 × 10^5^ cells/10 mL. The transformed protoplast yield at different concentrations of PEG-4000-CaCl_2_ apparently varied. It was revealed that adding 40% PEG-4000-CaCl_2_ obtained the highest transfection efficiency; the transformed protoplast yield reached about 5.0 × 10^5^ cells/10 mL (Figure 5A). Based on 40% PEG-4000-CaCl_2_, the transformation time was optimized to achieve the best transfection efficiency.

The results implied that 40% PEG-4000-mediated transfection time apparently affected transfection efficiency, and 30 min was optimal for the transformation of mango protoplasts, extending or shortening the transfection time would decrease transformed protoplast yield significantly (Figure 5B). Under florescence light, the green fluorescent protein (GFP) signal can be clearly observed in the protoplast. No green florescence protein signals were observed in the control treatment. The GFP signal of transfection using 40% PEG-4000-CaCl2 (Figure 5C) was observed after 20 min, and a stronger, larger GFP signal was observed after 30 min of transfection (Figure 5D) as the optimal transfection period. The GFP signal observed in the protoplasts showed that the exogenous gene (GFP gene) had been successfully introduced to the protoplast and that it was transcribed and translated effectively in the mango protoplast.

Most transformed protoplast cells (>80%) showed GFP fluorescence signals. This shows that protoplast-based transformation is a better way to study the transgene-encoded protein product at the molecular and biochemical levels. In addition, the tissue cell localized marker pC2300-35S vector containing GFP (pC2300-35S-GFP) was correctly targeted to the chloroplast membrane and nucleotide (Figure 6). *Phospholipase D1* (PLD1) plays a critical role in plant growth and development and signaling-dependent interactions. The practicability test showed that the *PLD1* gene fused to the GFP showed the presence of green fluorescence, verified via confocal microscopy (Figure 6B). As a positive control, the empty vector was transformed and observed under the microscope (Figure 6A). The subcellular localization of the chloroplasts here could be clearly observed by their red chlorophyll (Figure 6). The resulting plasmid was transformed into protoplasts, and after 30 min of incubation using 40% PEG-4000, the GFP signals were observed and clearly distinguishable (Figure 6A,B). In p35S-*PLD1*-GFP transformed cells, the green *PLD1*-GFP signal completely overlapped with the red auto-fluorescence located in the chloroplasts.

## 3. Discussion

Protoplast transient expression systems are a powerful tool for studying the molecular mechanisms underlying cell signaling processes. Isolating viable protoplasts from plant tissues for gene expression transformation is a key step in making a protoplast transformation method [9,19]. Here, we propose a simplified method for isolating leaf protoplasts and polyethylene glycol-calcium mediated transfection of *Mangifera indica* L. that can be used for gene function studies. Over the years, protoplast transient expression systems have been made for many plants, such as *Arabidopsis thaliana* [29], *Oryza sativa* [30], *Nicotiana tabacum* [31], and *Mangifera indica* L. [32]. This transformation system has become a fast and effective tool for transient expression assays, especially in plants that do not have an Agrobacterium-mediated transformation system [28]. Our reported transformed efficiency (>80%) in this paper was significantly improved in comparison to the transfection efficiency of mango protoplasts reported recently. The ability to separate a large number of viable protoplasts for high transfection efficiency and a large number of viable transfected cells will allow gene expression and environmental response studies to be conducted [33]. The type and concentration of the digestion solution used to isolate protoplasts from leaves were chosen based on the age and type of the leaves because a leaf that is too old has a lower yield of protoplasts [34].

Cellulase R-10, which is used to break down cell walls, is often used to separate protoplasts from *Triticum aestivum* [35] and *Catalpa bungei* [36] leaves. In *Chrysanthemum cv*., the protoplast yield was different when the cellulose in the digestion solution was increased from 0.5 g to 1.0 g [37]. Mannitol acts as an osmotic stabilizer to add osmolality to the digestion solution and keep the same amount of osmotic pressure on the inside and outside of the protoplast to keep it from breaking [32,38]. In this study, we found that 0.12 M and 0.04 M of mannitol reduced the protoplast yield and increased the survival rate compared to 0.16 M and 0.08 M of mannitol (Figure 4D). *Prunus avium* L. and *Carnation* cv. *Scarlet* protoplast cells maintained the balance between the interior and exterior osmotic pressures, preventing the protoplast’s loss and breaking by decreasing the mannitol concentration after transfection [9,32]. The digestion time also affects protoplast isolation, and this factor is dependent on the plant species type, which needs to be optimized empirically. Forming an appropriate digestion time for quality protoplasts was important, as long digestion times cause a decrease in the protoplast yield and viability rate [14]. In this study, we found that digestion times longer than 24 h resulted in a decrease in the protoplast yield and viability rate and that the optimal digestion time was 12 h. *Oryza sativa* maintained a stable yield after repeatedly digesting for 12 h [39]. These enzymes can decompose cellulose and other substances in the plant cell walls to release protoplasts. Owing to the different biological sources of these enzymes, they may have different optimal pH and temperature values. Based on the *Oryza sativa* leaf protoplast isolation system studies, the efficacy of combinations in the digestion solutions made significant differences in the protoplast yield [30]. Compared with other species, a significant increase in cellulase concentration increased protoplast yield [40]. From our studies, the most beneficial protoplast isolation conditions were cellulase R-10 (150 mg/mL) and macerozyme R-10 (180 mg/mL) at pH 5.6. The highest yield of leaf protoplasts was approximately 7.1 × 10^5^, which is different from *Torenia fournieri* leaves and showed that the highest yield of 6 × 10^4^ was obtained using cellulase R-10 (0.15 g) and macerozyme (0.05 g) in a 10 mL digestion solution digested for 4 h [41]. Similarly, in *Manihot esculenta Crantz*, the combination of 4.55 g/L mannitol, 0.1 g/L MES, 10 mg/L, and finally in 20 g/L) and a 16 h 1000 rpm shake gave the highest protoplast yield of 0.5 × 10^5^ [42]. Our study revealed a significant digestion compositing of the combination of two enzymes (macerozyme R-10 as cellulase R-10) and stabilizer, giving the highest protoplast cell yield without shaking for 12 h at an approximation of 7.1 × 10^5^ at 5.6 pH.

The pH value of the digestion solution indirectly affects the isolation of protoplasts by affecting enzyme activity. The protoplast yield and viability rate were highest when the pH was 5.6. In sweet cherry (*Prunus avium* L.), the protoplast yield decreased when the pH was increased above 5.8 [32]. This protocol provides a useful platform for gene expression and molecular network studies for a short period of time, which may have been previously difficult in mangoes. Recently, protoplast transient transformation and expression systems have been extended to studies on the dissection of protein-DNA binding [43] and expression [44]. As a result, when using DNA for genomic purposes, sensitivity reveals similar workable cells and tissue, with no significant difference between leaf tissue and a protoplast cell [45]. In our study, the majority (>80%) of transformed protoplast cells showed a GFP signal, indicating that protoplast-based transformation allows for more extensive molecular and biochemical analyses of the transgene-encoded protein product. Accordingly, the pC2300-35S transformed protoplasts with the right cellular context provided evidence of the GFP observed and merged, which confirmed the localization and expression in the cells. Generally, protoplasts rupture in hypotonic solution and collapse in hypertonic solution [46]. Under UV light, the GFP in the transformed protoplasts was observed, confirming that the target gene (*GFP* and *PLD1*) is transcribed and translated. Taken together, the leaf protoplast transient expression system established here is quick in supporting the general applicability of the transformation system.

This transient expression system enables the testing of multiple constructs, making it a powerful screening platform to test protein function, subcellular distribution, or function. As described in our studies, this method can be used to express proteins of interest that are fluorescently tagged and show how to make protoplasts from transiently expressing leaves to study how proteins of interest are distributed inside cells [7,11,47]. This present study can also be applicable to cellular functional studies and resistance studies that may also contribute to the damage to protoplast cells. In summary, the protocol presented here will enable in-depth studies of the molecular networks of the unique developmental regulatory mechanisms in mangoes.

## 4. Materials and Methods

### 4.1. Plant Materials

*Mangifera indica* L. cv. *Ketti* plants were grown in 10 × 10 cm pots and maintained in a semi-structured greenhouse under natural conditions. Young and healthy leaves were harvested and used for protoplast isolation.

### 4.2. Protoplast Isolation

Young open leaves were used for protoplast isolation. The leaves were cleaned by washing with 10% ethanol and cut into 0.5–1.0 mm strips using a fresh, sharp razor blade. The leaf strips were transferred to a Petri dish containing a freshly prepared digestion solution. The digestion solution was made as follows: cellulase R-10 (150 mg/mL), macerozyme R-10 (180 mg/mL), potassium chloride (KCl_2_) (20 mg/mL), and morpholin-4-ylethanesulfonic acid (MES) (3.88 mg/mL) at pH 5.6, diluted in double-distilled water and warmed up to 55 °C for 5–10 min. The digestion solution was allowed to cool to room temperature, after which 10 mg/mL calcium chloride (CaCl_2_) and 1 mg/mL bovine serum albumin (BSA) were added. The leaf strips were then completely submerged in the digestion solution and allowed to digest for 6, 8, 12, and 24 h in the dark. A total of 5 mg/mL of carbenicillin was added to avoid bacterial contamination. The pH value of digestion solution was determined to be 5.0, 5.2, 5.4, 5.6, 5.8, and 6.0 using a digital pH meter.

After the digestion period, the digestion solution and leaf strip mixture were gently agitated to release the protoplasts. The protoplast in the digested suspension was filtered through a 100µm cell strainer tube to remove tissue debris and centrifuged at 500× *g* for 6 min to pellet the protoplast cells. Before use, the cell strainer was kept in 95% ethanol and washed in water. The pelleted protoplast was dissolved in wash solution (WI) containing different amounts of mannitol (0.04, 0.08, 0.12, and 0.16 M), KCl_2_ (20 mg/mL), CaCl_2_ (10 mg/mL), BSA (1 mg/mL), and MES (40 mg/mL) at a pH of 5.6 and centrifuged again. The WI solution was carefully removed, and the cell pellet was resuspended in a pre-chilled suspension for counting (MMG) solution consisting of KCl_2_ (15 mg/mL), MES (40 mg/mL), and mannitol (0.04, 0.08, 0.12, and 0.16 M) at pH 5.6 to obtain isolated protoplasts for counting before transfection. The obtained protoplast was kept on ice for 30 min, after which the yield was counted using a hemocytometer under a microscope [48].

### 4.3. PEG-Calcium Preparation and Transformation Process

The protoplast concentration was adjusted to approximately 5.4 × 10^5^ cells/10 mL in MMG solution, and about 10 to 20 ug of the pC2300-GFP marker were mixed with a freshly prepared PEG-calcium chloride-base transfection solution. The PEG-calcium chloride-based transfection solution consisted of PEG-4000 (20, 30, 40, 50, 60, and 70%, *m/v*), mannitol (0.12 M), and CaCl_2_ (10 mg/mL) was prepared as follows: PEG-4000 and mannitol were firstly dissolved in water by heating it up to 60 ^0^ C for approximately 10–20 min. After the solution was cooled to room temperature, CaCl_2_ was added. The pC2300-35S-GFP in PEG-4000-CaCl_2_-protoplast solution was mixed gently and observed at different transformation optimal times (5, 10, 20, 30, 40, and 60) min at room temperature. The viability rate was determined using the count method [49]. The mixture was left in the dark for 16 to 20 h before the GFP signal in the transformed protoplast fluorescence images was photographed on a confocal microscope. Before being photographed, the transfected protoplast was washed in an MMG solution containing mannitol (0.04, 0.08, 0.12, and 0.16 M), KCl_2_ (15 mg/mL), and MES (40 mg/mL) (pH 5.6). and centrifuged at 200× *g* for 2 min. Then, the supernatant was carefully removed, leaving the transfected protoplasts. The excitation and emission wavelengths of the GFP were set as a major excitation peak at a wavelength of 395 nm and a minor one at 475 nm. Its emission peak was at 509 nm [50].

### 4.4. Protoplast Viability Test

A protoplast viability test was conducted to count the living and dead protoplast cells. A total of 0.4 g of trypan blue was dissolved in 80 mL of phosphate-buffered saline (PBS) and slowly came to a boil at pH 7.2. To a total volume of 100 mL, PBS was added after cooling to room temperature and keeping at room temperature to obtain the trypan blue standard solution. About 10 mL of protoplast cells were mixed in 0.1 mL of trypan blue standard solution. The mixture was loaded on a hemocytometer and examined instantly under a microscope for observation [51]. With a microscope and a hemocytometer, the protoplast viability of digestion time, mannitol concentration in WI solution, and MMG solution were seen, photographed, and counted [52]. After transfection, the mannitol concentration in the MMG wash solution (0.04, 0.08, 0.12, and 0.16 M) was also optimized in an after-transfection solution with KCl_2_ (15 mg/mL) and MES (40 mg/mL) (pH 5.6), and the viability rate was calculated. The viability rate was calculated as living protoplast number/total protoplast number × 100%.

### 4.5. Statistical Analysis

To determine the statistical significance of differences between sample means, multiple pairwise comparisons (*p* < 0.05) were performed. The efficiency and optimal values were validated based on the observed factors: mannitol concentration of the digestion solution, WI solution and MMG solution, digestion time, pH value, polyethylene glycol (PEG) concentration, and transfection time.

## 5. Conclusions

This study developed an efficient protoplast isolation and transformation system, which can help a lot in the gene expression study of mangoes. The protoplast isolated yield and efficiency were dependent on the specific digestion enzyme solution. Leaf tissue digested in digestion solution consisting of cellulase R-10 (150 mg/mL), macerozyme R-10 (180 mg/mL) for 12 h, and then in WI and MMG solution with 0.16 and 0.08 mannitol, respectively, will obtain the highest protoplast yield and protoplast viability. The most beneficial protoplast transformation condition was mediated by 40% calcium-chloride-based polyethylene glycol (PEG)-4000-CaCl_2_ for 30 min. The GFP signal was observed, confirming that the target gene was transcribed and translated.

## Figures and Tables

**Figure 1 ijms-24-11984-f001:**
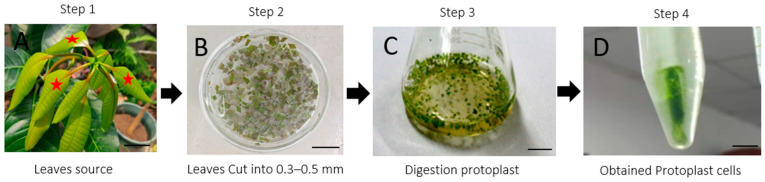
Overview of the experimental procedure for *Mangifera indica* L. protoplast isolation. Step 1: Preparation of fresh, young, healthy, and fully expanded leaves (indicated as red star) (**A**). Step 2: Cutting and submerging the leaf segments into the digestion solution (**B**). Step 3: The released protoplast in the digestion solution (**C**). Step 4: The protoplast observed in the washing solution (**D**).

**Figure 2 ijms-24-11984-f002:**
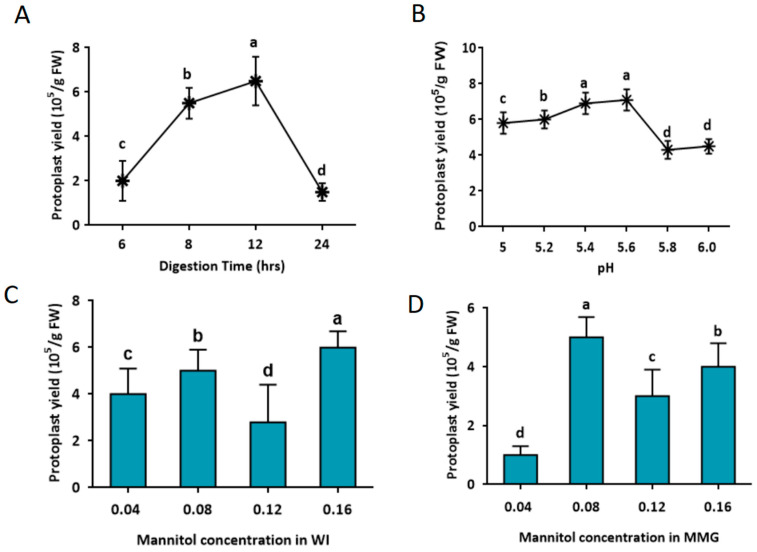
Optimizing conditions for protoplast isolation. Effects of digestion time on protoplast yield (**A**), effects of pH value of the digestion solution on protoplast yield (**B**), effects of mannitol concentration in washing solution (WI) on protoplast yield (**C**), and effects of mannitol concentration in suspension for counting solution (MMG) on protoplast yield (**D**). The different lowercase letters labeled above the columns indicate a significant difference at *p* ≤ 0.05 between the columns.

**Figure 3 ijms-24-11984-f003:**
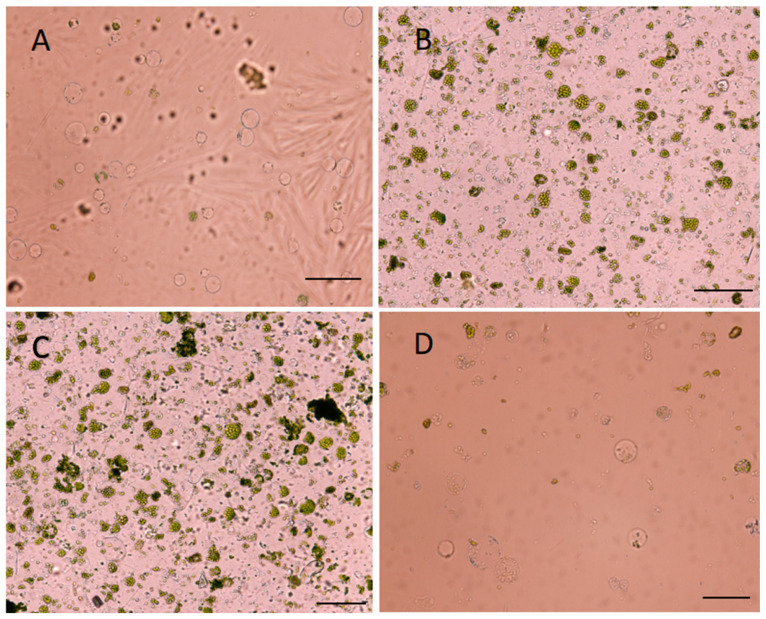
Protoplasts isolated from ‘Ketti’ leaf tissue. The protoplasts were produced, separated at the interface, and purified using a 100 µm strainer. The protoplast yield was observed after hours of digestion after 6 h of digestion (**A**), after 8 h of digestion (**B**), after 12 h of digestion (**C**), and after 24 h of digestion (**D**). Scale bars = 100 µm.

**Figure 4 ijms-24-11984-f004:**
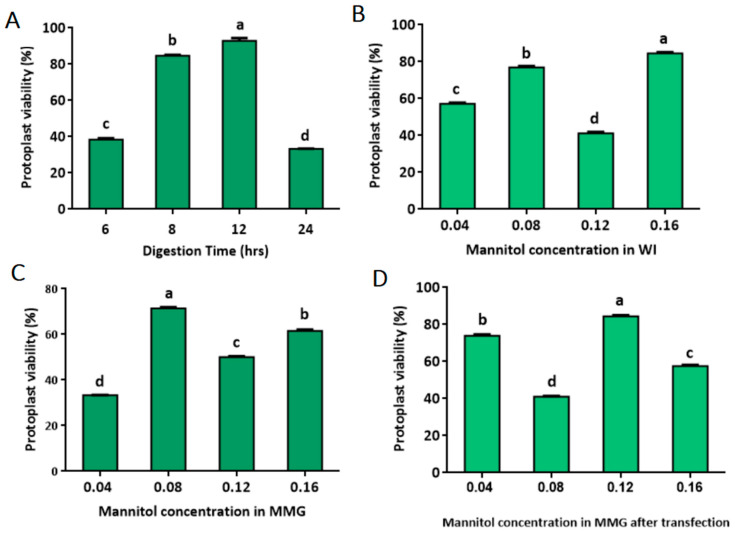
The mannitol concentration on protoplast viability. (**A**) Effect of digestion time on protoplast viability rate; (**B**) effect of mannitol concentration in WI solution on protoplast viability rate; (**C**) effect of mannitol concentration in MMG solution on protoplast viability rate; (**D**) effect of mannitol concentration in suspension for counting MMG solution after transfection. Different lowercase letters labeled above the columns indicate a significant difference at *p* ≤ 0.05 between the columns.

**Figure 5 ijms-24-11984-f005:**
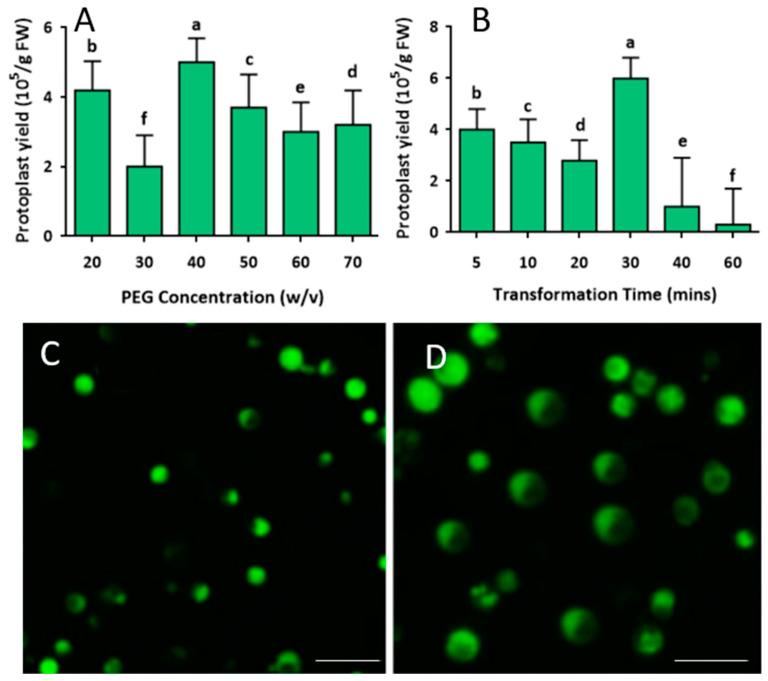
Protoplast transfection condition optimization. (**A**) Effect of PEG-4000-CaCl_2_ concentration on protoplast transformation efficiency; (**B**) effect of transfection time on protoplast transfection efficiency; (**C**) GFP signal observation of transformed protoplast after 20 min of transfection; (**D**) GFP signal observation of transformed protoplast after 30 min of transfection. Different lowercase letters labeled above the columns indicate a significant difference at *p* ≤ 0.05 between the columns. Scale bars = 100 µm.

**Figure 6 ijms-24-11984-f006:**
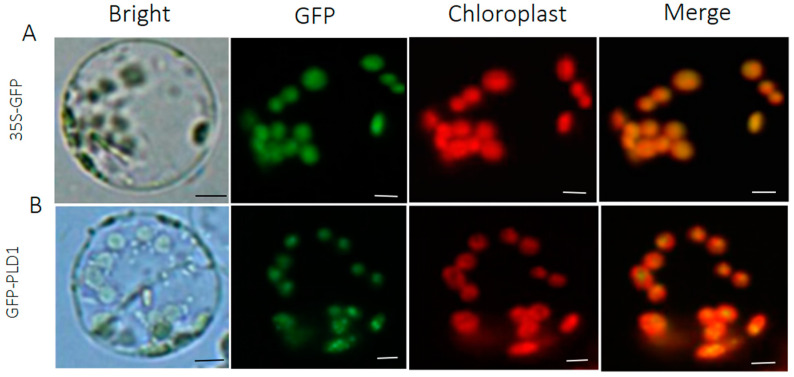
Subcellular localization of the GFP signal in the transformed mesophyll protoplast. (**A**) The *p35S-GFP* signal (green) completely localized with chloroplast (red) overlapped under auto-fluorescence merging and (**B**) the *p35S-PLD1-GFP* signal is completely localized with chloroplast auto-fluorescence. Scale bars = 100 µm.

## Data Availability

Not applicable.

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
