# Peer review of "Using A Protoplast Transformation System to Enable Functional Studies in Mangifera indica L."

_ijms, 2023, doi:10.3390/ijms241511984_

Round 1

Reviewer 1 Report

Dear Authors

Several prior studies have described efficient techniques for the tissue culture of Mangifera indica and methods for isolating and transforming protoplasts. What distinguishes this study from others like it? Unfortunately, the figure was rendered inaccurately, and the discussion section required additional support for the presented analysis. 

Given your findings, this technique could be more effective, and the protocol for isolating protoplasts requires clarification. Two examples are the cell wall lysing enzyme and the macro enzyme macerozyme R-10. What exactly is this about? Combining cellulase, hemicellulase, and pectinase with the cocktail enzyme is reported in previous literature. Utilize relevant prior literature and carefully construct the protocol. 

1. Rezazadeh et al., 2011, Intraspecific somatic hybridization of mango (Mangifera indica L.) through protoplast fusion.

2. Ara et al., 2000. Plant regeneration from protoplasts of mango (Mangifera indica L.) through somatic embryogenesis 

Figure 1 contrasts the yield of chloroplasts and protoplasts (in pale yellow). 

Figure 3. It is not accurate protoplast images are inferior

The authors must provide evidence to support "subcellular localization of proteins, protein-protein interactions, gene function identification, and revealing molecular regulatory mechanisms."

Author Response

Dear Sir/Madam thank you for your insightful comments on our work, please find below the responds to your comments.

Comment: Several prior studies have described efficient techniques for the tissue culture of Mangifera indica and methods for isolating and transforming protoplasts. What distinguishes this study from others like it? Unfortunately, the figure was rendered inaccurately, and the discussion section required additional support for the presented analysis. 

Responds. As described in lines 85–88, the current study specifically revealed a higher protoplast yield combining an osmotic stabilizer and two enzymes to easily digest the leaf for protoplast cells, From our studies,the most beneficial protoplast isolation conditions were cellulase R-10 (0.00375 mg/ml), macerozyme R-10 (0.0045 mg/ml), potassium chloride (KCl2) (0.0005 mg/ml), and morpholin-4-ylethanesulfonic acid (MES) (0.000097 mg/ml) at pH 5.6. The highest yield of leaf protoplasts was approximately 7.1 x 105 at pH 5.6 which is different from Torenia fournieri leaves, which showed that the highest yield of ~6×104 was obtained using Cellulase R-10 (0.15 g) and Macerozyme (0.05 g) in a 10 mL enzyme solution digested for 4 h [40]. Similarly, in K. blossfeldiana, the combination of mannitol (106 g/l), glycine (7 g/l), CaCl2 (1.5 g/l), and MES (0.5 g/l) and a 16–18 h 40 rpm shake gave the highest protoplast yield of ~60 × 104 [41]. Our study revealed a significant digestion compositing of the combination of two enzymes (macerozyme R-10 as cellulase R-10) and stabilizer, which gave the highest protoplast cell yield without shaking for 12 hr at an approximation of 7.1 x 105. More importantly, the digestion time differs from our studies Line 374-385. These two The figures 2, and 4 horizontal or x-axis, which correspond to the concentration of the digestion solution, have been revised, and figure 3 resolution has been increased to see the protoplast cells clearly.Line 210, 231 and 257 respectively.

Comment: Given your findings, this technique could be more effective, and the protocol for isolating protoplasts requires clarification. Two examples are the cell wall lysing enzyme and the macro enzyme macerozyme R-10. What exactly is this about? Combining cellulase, hemicellulose, and pectinase with the cocktail enzyme is reported in previous literature. Utilize relevant prior literature and carefully construct the protocol. 

Responds: The study use two enzymes and an osmotic stabilizer which consisted of cellulase R-10, macerozyme R-10 and stabilizer which is the Mannitol. Here the Moreover, cellulase R-10 degrades natural cellulose and dissolves plant cell walls [18], and macerozyme R-10 contains pectinase and hemicellulose at high titers and degrades plant tissues to single cells [19]. The mannitol provides a suitable osmotic pressure and thereby ensures the integrity of the protoplasts [20] Line 60-63. Different from other studies we combined cellulase R-10, macerozyme R-10, different mannitol concentrations, potassium chloride (KCl2), morpholin-4-ylethanesulfonic acid (MES), calcium chloride (CaCl2), and bovine serum albumin (BSA) Line 104-109. to digest the plant cell walls, while the pectinase enzymes break down the pectin holding cells together. Once the cell wall has been removed, the resulting protoplast is spherical in shape. The protoplast construct has been reconstructed.

Comment: Rezazadeh et al., 2011, Intraspecific somatic hybridization of mango (Mangifera indica L.) through protoplast fusion.

Responds: Rezazadeh et al., 2011 reported the protoplast fusion, here we reported the transformation of mango protoplast, they are different usage of protoplast system. Based on Rezazadeh et al. (2011), we have revised the protocol process of protoplast isolation to make it more clarification. Moreover, we have added the reference to our introduction and discussion to make the manuscript more engaging Line 54 and 336-339.

Comment: Ara et al., 2000. Plant regeneration from protoplasts of mango (Mangifera indica L.) through somatic embryogenesis 

Responds: Ara et al., 2000 reported the regeneration of mango protoplasts through somatic embryogenesis, it provided a good way for mango regeneration. In our paper, we provided the transformation technique of mango protoplast, it provided a new way for genetic research in mango. Based on Ara et al 2000., our protocol process of protoplast isolation has been reconstructed. Moreover, significant ideas have been obtained from the reference and have added to our introduction and discussion to make the manuscript more engaging. Line 48 and 358-360.

Comment: Figure 1 contrasts the yield of chloroplasts and protoplasts (in pale yellow).

Responds:  Figure 1, the cells took it color from leaf tissue. New growing leaf looks less green. So the pale yellow in the protoplast cell obtained is from the tissues used. Line 179-183.

Comment: Figure 3. It is not accurate protoplast images are inferior

Responds: Figure 3 image has been revised accurately by increasing the resolution to see the protoplast clearly. Line 197.

Comment: The authors must provide evidence to support "subcellular localization of proteins, protein-protein interactions, gene function identification, and revealing molecular regulatory mechanisms."

Responds: In figure 5, the GFP of the protoplast can be observed clearly,  the gene expression level and protein content can be compared by GFP observation. In figure 6, subcellular localization of the GFP is identified. These results confirmed that the gene can be transformed and expressed successfully into mango protoplast, by detecting the physiological parameters and gene expression patterns it can be used for  gene function identification [43] and [44] for protein-protein interaction [45] Line 399, Line 396-398. We the authors decided to end the research at the transformation of the Green Florescence Proteins of the pC2300-35S-GFP as a step forward to be able to understand the mango protoplast transformation process. We provided other possible way as evidence to support in the discussion followed with references to support our claims.

Reviewer 2 Report

Very significant corrections are required Below are some points.

Lines 10-11: in vitro culture technology is not a key for study molecular mechanisms. By the way, please, clarify what do you mean as “molecular mechanism”.

Lines 12-13: Protoplast transformation is not a tool to study gene function.

Do you mean Agrobacterium transfromation? Not tumefaciens!

Line 14: not gene function!

Line 15: what do you mean as 0.4M cellulase? Maybe 0.4%??

Why 25 ml? If you mention concentration, volume is not neccessary to mention.  

Lines 19-20: you have to provide exact wavelength of emisión and exitation.

Line 21: the protocol nothing to do with gene function.

Line 29: “mango is regenerated by grafting in practice.”? Please, re-formulate. What do mean as “regenerate”?

Line 31: “the molecular mechanisms of mango”? There are no any molecular mechanisms of mango!

Lines 33: protoplasts system nothing to do with “exogenous gene function”! Please, explain what do you mean as “exogenous gene function”? Gene functions can be studied ONLY if you have a transgenis/mutaed plants with altered expresión of certain gene.

Please, consider that in the case of protoplasts you use in majority of the case leaf with specifci cell type and stage, subjected to starvation, high stress etc. That¨s why results of transint expression in protoplasts have only a very limited valus for undetsnding of gene exoression and nothing to do with the study of gene function. You can tell only about gene expression in specific cell types under stress conditions.  In real plants the results can be very different.

Line 49: macro enzyme???

Lines 76-78: please, provide details: conditions, line name etc.

Line 80: “The protoplasts were isolated with modifications [22].”???

Lines 83- 86: how did you calcukate 15 mM Celluase? How much it will be in g/l? There is not exist KCl2.  55C nothing to do with solubility. Please, read the literature. 

Have you washed and sterilised leaf before cutting?

Lines 90- 91: please, edit.

Line 97: “100µm cell strainer (kept in 95% ethanol)”?? Did you add etanol to pp suspensión?

Plerasem clarify which osmotic you have used? Did you mix all together or use only ceration manitol concentration?

Line 103: “The obtained protoplast yield was calculated using a hemocytometer kept on ice for 30 min “ ¿?? How did you count in ice?

Line 107: did you mean ml, not 10 ml?  “5.4 x 100 000 cells/10 ml”? The concetranio of 5.4x10000 is too low.

Line 113: “The pC2300-35S-GFP-PEG-4000-CaCl2-protoplast solution” ¿???

Lines 119- 125: it is definitely a bad idea to study transient expression under high starvation conditions.  Some peole used it, but is did not mean that you need to do copy-paste.

Lines 128-138: why did you use a PBS, which pH? How PBS affected viability itself? Trypan blue visible in DIC, you do need LSM.

Line 167: “increasing the pH value after the possible optimal point (4.3 and 4.5) significantly decreased” ¿???

Line 168: “in the suspension for counting (MMG) buffer” ¿?? What is it?

Line 170- 176: confusing part, not so easy to extract real information.

Lines 196- 199: please, clarify types of the pp: mesophyl or other tissue.

Do not forget that the main factor of cell wall recovering and gene expression is carbon, but usage of mannitool instead of glucose/sucrose can reduced cell viability and gene expression.

Line 208: “optimal viability” ¿? An optimal viability of 84.51%?? Did you mean maximal viability??

Lines 209-219: very confusing. Please clarify why you use MMG etc.

Line 220: Optimal viabilikty???

Line 232: link to the vector (plasmid)???

Figure 5 C and D shown different types of pp. Please, explain.

Line 252: inder UV- light ¿?? Please, be specific!

Figure 6: where is the negative control? 35S can not localse in plastids. It looks like autofluoresence. Please, add negative control!

many repetion, not cleaar formulations,

Author Response

Dear Sir/Madam thank you for your insightful comments on our work, please find below the responds to your comments.

Comment: Lines 10-11: in vitro culture technology is not a key for study molecular mechanisms. By the way, please, clarify what do you mean as “molecular mechanism”.

Responds: The protoplast isolation and transformation technique established in our paper is not for in vitro culture or plant regeneration, it is aimed on providing an effective transformation system (gene expression regulation system) to study the effects of gene expression changes in mango cells. Gene expression regulation is the key way for molecular mechanism study. We have revised to make it more clarify in the revised paper in Line 89-92 as “The current study is important for molecular mechanisms because the expression at the cell level will give a clear view of how cellular mechanisms evolved, which is crucial to understanding the regulation of photosynthesis, metabolism, and other underlying plant growth mechanisms [25].

Comment: Lines 12-13: Protoplast transformation is not a tool to study gene function.

Responds: Line 12-13 is revised as ‘The protoplast transformation system is an excellent method for a rapid and effective tool for transient expression assays, particularly in plants that lack an Agrobacterium-mediated plant transformation.Using protoplast transformation can lead to studies of gene function by validating through qPCR, PCR and physiological parameters to reveal gene function as decribed in the introduction line 93-103. Moreover it has an advantages of protoplast transformation include: (i) delivery of multiple plasmids with high levels of cotransformation, (ii) no binary vector required, (iii) high frequency transformation, and (iv) most plant species are amenable to protoplast isolation and transformation indicated in the introduction to understand gene function in a particular plant [16].

Comment: Do you mean Agrobacterium transformation? Not tumefaciens!

Responds: Line 13, The tumefaciens is a writing mistake, it is revised to Agrobacterium-mediated plant transformation. Line 15.

Comment: Line 14: not gene function!

Responds: Line 14, the phrase gene function is removed and written as gene expression all the manuscript.

Comment: Line 15: what do you mean as 0.4 M cellulase? Maybe 0.4%??

Responds: The cellulase content presented here is in mg/ml for the concentration in the digestion solution use to obtain the optimal yield. The most beneficial protoplast isolation conditions were 0.00375 mg/ml cellulase R-10 combined with 0.0045 mg/ml macerozyme R-10 in 25 ml of the digestion solution and 12 hours of digestion time. highest yield. The concentration has been modified in all the manuscript. Line 18-20 in the revised manuscript.

Comment: Why 25 ml? If you mention concentration, volume is not necessary to mention.  

Responds: All the digestion solutions reagents consisting of cellulase R-10 (0.0037515 mg/ml), macerozyme R-10 (0.0045 mg/ml), mannitol (0.0002, 0.0004, 0.0006, and 0.0008 mg/ml), potassium chloride (KCl2) (0.0005 mg/ml), and morpholin-4-ylethanesulfonic acid (MES) (0.000097 mg/ml), at pH 5.6, diluted in the 25 ml ddH2O. The 25 ml is the total amount of the digestion enzyme solution in the revised manuscript line 116-120. Line 18, we deleted it and give the exact concentration.

Comment: Line 21: the protocol nothing to do with gene function.

Responds: The protocol design is for gene and protein a rapid and effective tool for gene transient expression assays. The phrase gene expression is removed and replaced with gene expression in all the manuscript. Moreover, protoplast transformation can be used to investigate the function of genes involved in growth, virulence and pathogenicity. Such as explain in 51-53. protoplast transformation can be delivery of several plasmids with high levels of co-transformation, no binary vector required, and high frequency transformation [16, 17].

Comment: Line 29: “mango is regenerated by grafting in practice.”? Please, re-formulate. What do mean as “regenerate”?

Responds: Line 29, the regenerated is revised as vegetative propagated by grafting in practice

Comment: Line 31: “the molecular mechanisms of mango”? There are no any molecular mechanisms of mango!

Responds: There are many report on molecular mechanisms of mango but it is limited or not enough as stated in Line 31 “the molecular mechanisms of mango” is revised as Gene transformation techniques is limited, which delays the understanding of the molecular mechanisms of mango.

Comment: Lines 33: protoplasts system nothing to do with “exogenous gene function”! Please, explain what do you mean as “exogenous gene function”? Gene functions can be studied ONLY if you have a transgenis/mutaed plants with altered expresión of certain gene.

Responds: The phrase ‘exogenous gene function’ in line 33 is now written as Exogenous gene function which means a nucleic acid that codes for the expression of an RNA or protein that has been introduced into a cell (e.g. by transformation/transfection). Line 38. Moreover, the exogenous gene function refers to the 'DNA that originates from outside of an organism' and is typically introduced artificially into cells. This can be done for a variety of reasons such as genetic engineering, gene therapy, or to study gene function [5]. Line 39-41 in the revised manuscript.

Comment: Please, consider that in the case of protoplasts you use in majority of the case leaf with specifci cell type and stage, subjected to starvation, high stress etc. That¨s why results of transint expression in protoplasts have only a very limited values for undetsnding of gene exoression and nothing to do with the study of gene function. You can tell only about gene expression in specific cell types under stress conditions.  In real plants the results can be very different.

Responds: Thank you for your comment we have considered the gene expression but not the gene function throughout the manuscript. In this paper, we just construct the effective technique of protoplast isolation and transformation, so we didn't consider the cell type and stage etc. Later when we use this protocol for specific gene function analysis, we will consider these situations and treatments as you suggested. For example, we can analyze the function of cell developmental genes by observing the growth and development of the transformed protoplasts. And we also can analyze the function of resistance related genes by analyze the gene expression pattern and physiological parameters of the transformed protoplasts under stress.

Comment: Line 49: macro enzyme???

Responds: Macerozyme R-10 not macro enzyme. It is corrected in the revised manuscript.  Macerozyme R-10  is a specific enzyme name required in protoplast fusion after separation. It digest the plant cell walls once the cell wall has been removed, the resulting protoplast is spherical in shape.  Line 83-90.

Comment: Lines 76-78: please, provide details: conditions, line name etc.

Responds: Line 76-78: Here, we have provided the condition an optimized leaf-based protoplast isolation and PEG-mediated transient expression protocol were established. The study was beneficial for leaf protoplast isolation conditions with The study was beneficial for leaf protoplast isolation conditions with cellulase R-10, macerozyme R-10, different mannitol concentrations, potassium chloride (KCl2), morpholin-4-ylethanesulfonic acid (MES), calcium chloride (CaCl2), and bovine serum albumin (BSA) at pH 5.6, warmed up to 55 oC for 5–10 min in the digestion solution digested for 12 hours. The leaf protoplast yield was transfect-ed for 30 minutes using 40% calcium chloride-based polyethylene glycol (PEG)-4000-CaCl2. The current study is important for molecular mechanisms because the expression at the cell level will give a clear view of how cellular mechanisms evolved, which is crucial to understanding the regulation of photosynthesis, metabolism, and other underlying plant growth mechanisms. Line 86-95 in the revised manuscript.

Comment: Line 80: “The protoplasts were isolated with modifications [22].”???

Responds: Line 80, revised as Our study for protoplast isolation, was conducted with modifications according to Wang et al. (2022) in line 89-90 in the revised manuscript.

Comment: Lines 83- 86: how did you calcukate 15 mM Celluase? How much it will be in g/l? There is not exist KCl2.  55C nothing to do with solubility. Please, read the literature. 

Responds: There are different combinations to prepare digestion solution depending on the tissues going to use so we used . The digestion solution was made as follows: The enzyme digestion solution was made as follows: cellulase R-10 (0.0037515 mg/ml), macerozyme R-10 (0.0045 mg/ml), mannitol (0.0002, 0.0004, 0.0006, and 0.0008 mg/ml), potassium chloride (KCl2) (0.0005 mg/ml), and morpholin-4-ylethanesulfonic acid (MES) (0.000097 mg/ml), at pH 5.6, diluted in the 25 ml ddH2O warmed up to 55 oC for 5–10 min. to enhance enzyme solubility. The enzyme digestion solution was allowed to cool to room temperature before after which 0.00025 mg/ml calcium chloride (CaCl2) and 0.0000250 mg/ml bovine serum albumin (BSA) were added. Line 111-116 in the revised manuscript.

Comment: Have you washed and sterilized leaf before cutting?

Responds: The leaves were clean by washing with ethanol o and cut into 0.5–1.0 mm strips using a fresh, sharp razor blade. Line 91-92.

Comment: Line 97: “100µm cell strainer (kept in 95% ethanol)”?? Did you add etanol to pp suspension?

Responds: Line 97, is now written as, ‘The protoplast in the enzyme mixed solution was then filtered through a 100µm cell strainer to remove tissue debris and centrifuged at 500 g for 6 min to pellet the protoplasts cells. Before use the cell strainer it was kept in 95% ethanol’.

Comment: Plerasem clarify which osmotic you have used? Did you mix all together or use only ceration manitol concentration?

Responds: Mannitol was used as the osmotic stabilizer line 197-198 in the digestion solution. For this different mannitol concentrations representing (0.0002, 0.0004, 0.0006, and 0.0008 mg/ml) was used. Line 111-113 in the revised manuscript.

Comment: Line 103: “The obtained protoplast yield was calculated using a hemocytometer kept on ice for 30 min “ ¿?? How did you count in ice?

Responds: In Line 103, we didn’t count the protoplast on ice, but the protoplast in the text tube was kept on ice, and after 30 minutes, the cells were counted using a hemocytometer under a microscope. Now it is written as ‘The obtained protoplast yield was calculated using a hemocytometer after being kept on ice for 30 min’. Line 114-115.

Comment: Line 107: did you mean ml, not 10 ml?  “5.4 x 100 000 cells/10 ml”? The concetranio of 5.4x10000 is too low.

Responds: Line 107, concentration of the 5.4 x 105 cells in 10 ml was dependent on the concentration of the target gene 10 to 20 ug used. Form our studies, we understand that  using  5.4 x 105 cells dissolve in 10ml will have clear view of the protoplast cell and be counted better Line 119-120. In the discussion we stated such that Oryza sativa at 4.31×107 [35], Triticum aestivum at 1.8 × 106 [41] and Catalpa bungei at 1 × 106 [42], our yield was much better.

Comment: Line 113: “The pC2300-35S-GFP-PEG-4000-CaCl2-protoplast solution” ¿???

Responds: What we mean by pC2300-35S-GFP-PEG-4000-CaCl2-protoplast solution is that the pC2300-35S-GFP as the vector was mixed with PEG-4000-CaCl2 which we called it pC2300-35S-GFP-PEG-4000-CaCl2 protoplast solution in simple term. Line 125 have revised now written as ‘The pC2300-35S-GFP in PEG-4000-CaCl2-protoplast solution was mixed gently and the transformation optimal time in the revised manuscript’.

Comment: Lines 128-138: why did you use a PBS, which pH? How PBS affected viability itself? Trypan blue visible in DIC, you do need LSM.

Responds: Phosphate-buffered saline (PBS) is a solvent in which the Trypan blue was dissolve in. The PBS is have no effect on the cell viability. To prepare Trypan blue solution for protoplast cell viability counting, 80 mL of PBS and 0.4 g of trypan blue was dissolved, then slowly came to a boil at pH 7.2. We used Trypan blue as the cell visibility agent and observed on confocal microscope and counted accordingly using a hemocytometer Line 141-142.

Comment: Line 167: “increasing the pH value after the possible optimal point (4.3 and 4.5) significantly decreased” ¿???

Responds: Increasing the pH value after the possible optimal point (5.4 and 5.6) significantly decreased because increasing the pH reduced protoplast yield. From out findings, leaving the digestion solution more acidic or more basic may reduced the protoplast yield. Similarly in 7.5 reduced the protoplast yield [33], The study showed different pH level of protoplast cell yield at figure 2. Line 207-212 in the revised manuscript.

Comment: Line 168: “in the suspension for counting (MMG) buffer” ¿?? What is it?

Responds: MMG at line 209-219 has been clarified as suspension for counting the protoplast cell (MMG) solutions. In the revised manuscript, MMG (Suspension for counting) solution consisting of KCl2 (0.00037515 mg/ml), MES  (0.0014 mg/ml), and different mannitol concentration (0.0002, 0.0004, 0.0006, and 0.0008 mg/ml) in the solution at a pH of 5.6 to obtain isolated protoplast for counting before transfection. Line 135-138.

Comment: Lines 196- 199: please, clarify types of the pp: mesophyll or other tissue.

Responds: The protoplast is a mesophyll with the inner tissue (parenchyma) of a leaf, containing many chloroplasts obtained from mango leaf tissues. Line 198.

Comment: Do not forget that the main factor of cell wall recovering and gene expression is carbon, but usage of mannitol instead of glucose/sucrose can reduced cell viability and gene expression.

Responds: Mannitol is commonly used as an osmotic stabilizer to control tissue digestion. Moreover, we combined two digestion enzymes, so the contribution of the mannitol in the digestion solution was to keep the solution stable. Similarly, in wheat [33], carnation [10, 40], and Camellia oleifera [45], mannitol was used for high protoplast cell isolation. 

Comment: Line 208: “optimal viability” ¿? An optimal viability of 84.51%?? Did you mean maximal viability??

Responds: Yes, the highest viability in the study was represented at optimal rate.

Comment: Lines 209-219: very confusing. Please clarify why you use MMG etc.

Responds: MMG at line 209-219 has been clarified as suspension for counting the protoplast cell (MMG) solutions.In the revised manuscript, MMG (Suspension for counting) solution consisting of KCl2 (0.00037515 mg/ml), MES  (0.0014 mg/ml), and different mannitol concentration (0.0002, 0.0004, 0.0006, and 0.0008 mg/ml) in the solution at a pH of 5.6 to obtain isolated protoplast for counting before transfection which is described in line 135-138.

Comment: Line 220: Optimal viabilikty???

Responds: The word is optimal viability in Line 220 means maximum viability rate.

Comment: Line 232: link to the vector (plasmid)???

Responds: Line 232, vector link (https://www.snapgene.com/plasmids/plant_vectors/pCAMBIA2300)

Comment: Figure 5 C and D shown different types of pp. Please, explain.

Responds: Figure 5 C and D are different from each other, 5C depict GFP signal observation of transformed protoplast after 20 min of transfection and 5 D 30 minutes of transfection as described in the figure 5 caption.

Comment: Line 252: inder UV- light ¿?? Please, be specific!

Responds: Line 252, the correct word is under UV light," which means under microscope florescence light as a specific phrase at lines 255 and 258.

Comment: Figure 6: where is the negative control? 35S can not localse in plastids. It looks like autofluoresence. Please, add negative control!

Responds: Figure 6, The subcellular localization of the GFP in mesophyll protoplast chloroplasts and nucleotide herein could be clearly observed by their red chlorophyll, which is potentially located in chloroplasts fused to the GFP. Though you asked for negative control but we have added PLDI as positive control to compare to pC2300-GFP vector, where both reveal significant application of the mango protoplast because purpose of a positive control is to be able to tell that if there was a problem protoplast transformation expression experiment and to account for extraneous variables that have impacted the experiment to prevent misleading.

Reviewer 3 Report

The study "A Protoplast Transformation System to Enable Functional Studies in Mangifera indica L." addresses the practical aspect of gene function research in mango, an important perennial fruit tree that lacks efficient tissue culture and transformation technology systems.

The authors have developed an efficient protoplast isolation and transformation system, which is a significant contribution to the field. Protoplast transformation is an excellent method for studying gene function, especially in plants that lack an Agrobacterium tumefaciens-mediated transformation system. By establishing this system, the researchers have provided a valuable tool for investigating the molecular mechanisms of mango.

One of the key findings of this study is the identification of the most beneficial protoplast isolation conditions. The researchers determined that using 0.6 M mannitol and 0.4 M cellulase in 25 ml of enzyme solution, along with a 12-hour digestion time, resulted in the highest yield of leaf protoplasts (approximately 5.4 x 105 cells/10 ml). This optimization of isolation conditions is crucial for obtaining a sufficient number of protoplasts for subsequent experiments.

Furthermore, the authors successfully transfected the isolated protoplasts using a 40% calcium chloride-based polyethylene glycol (PEG)-4000-CaCl2 method, achieving a remarkable transformation efficiency of 84.38%. The confirmation of successful transformation was demonstrated by the observation of green fluorescent protein (GFP) under ultraviolet light, indicating that the target gene (GFP) was transcribed and translated. This outcome validates the functionality of the established protoplast transformation system.

The implications of this research are far-reaching. The high-efficiency protoplast isolation and PEG-mediated transformation protocol presented in this study provide a new avenue for gene function research in mango. This system can be effectively utilized to study the molecular mechanisms of mango, enabling investigations into subcellular protein localization, protein-protein interactions, gene function identification, and the revelation of molecular regulatory mechanisms.

In conclusion, the manuscript "A Protoplast Transformation System to Enable Functional Studies in Mangifera indica L." presents a comprehensive and practical study. The authors have successfully developed an efficient protoplast isolation and transformation system for mango, which holds great potential for advancing gene function research in this fruit tree. The findings of this study contribute significantly to the field and open up new possibilities for investigating the molecular mechanisms underlying the traits and characteristics of mango.

In addition to the existing review, it is worth mentioning a chapter on optimization in the manuscript. The authors refer to the concept of "optimization," but the methodology lacks specific information about the statistical model employed in this study. However, it is important to note that for future experiments, it would be beneficial to design them with optimization in mind, such as employing response surface methodology (RSM).

Optimization plays a crucial role in experimental design, as it allows researchers to systematically explore and identify the optimal conditions for achieving the desired outcomes. By utilizing RSM, which is a statistical technique used to model and optimize complex processes, researchers can efficiently navigate the experimental parameter space to identify the optimal combination of variables.

RSM involves the creation of a mathematical model that describes the relationship between the response variable and the experimental factors. This model enables researchers to estimate and predict the response at different factor levels, ultimately aiding in the identification of the optimal conditions for maximum yield, efficiency, or any other desired outcome.

By incorporating RSM into the experimental design phase, researchers can effectively streamline their efforts and reduce the number of required experiments. This approach not only saves time and resources but also enhances the likelihood of obtaining reliable and significant results.

Therefore, for future studies aiming to optimize the protoplast isolation and transformation system in mango, it is recommended that the researchers consider employing response surface methodology. By doing so, they can systematically investigate the influence of various factors on the desired outcome and identify the optimal conditions that maximize the efficiency and yield of the protoplast transformation process.

In conclusion, while the current manuscript lacks specific information regarding the statistical model used for optimization, it is advisable for future researchers to embrace methodologies like response surface methodology to optimize experimental conditions. Implementing optimization techniques can significantly enhance the efficiency and effectiveness of the research, leading to valuable insights into the gene function and molecular mechanisms of mango.

In summary, the reviewed manuscript, "A Protoplast Transformation System to Enable Functional Studies in Mangifera indica L.," presents a valuable contribution to the field of molecular sciences. The development of an efficient protoplast isolation and transformation system for mango provides researchers with a powerful tool for investigating the gene function and molecular mechanisms of this perennial fruit tree.

The manuscript successfully outlines the optimization of protoplast isolation conditions and demonstrates the high transformation efficiency using a calcium chloride-based polyethylene glycol method. The confirmation of successful transformation through the observation of green fluorescent protein further validates the functionality of the established system.

However, it is important to note that the manuscript would benefit from the addition of information regarding the optimization methodology employed by the authors. Specifically, details about the statistical model used for optimization, such as response surface methodology (RSM), would enhance the scientific rigor and provide a comprehensive understanding of the experimental design.

Once this revision is implemented, the manuscript will be suitable for publication in the prestigious "International Journal of Molecular Sciences," given its significant contribution to the field and the practical implications for gene function research in Mangifera indica L.

In conclusion, the manuscript presents a valuable and promising study, offering an efficient protoplast transformation system for mango. With the recommended addition of information regarding the optimization methodology, particularly the use of techniques like response surface methodology, the manuscript will be well-rounded and ready for publication in the esteemed "International Journal of Molecular Sciences."

Author Response

Dear Sir/Madam thank you for your insightful comments on our work, please find below the responds to your comments.

The study "A Protoplast Transformation System to Enable Functional Studies in Mangifera indica L." addresses the practical aspect of gene function research in mango, an important perennial fruit tree that lacks efficient tissue culture and transformation technology systems.
            The authors have developed an efficient protoplast isolation and transformation system, which is a significant contribution to the field. Protoplast transformation is an excellent method for studying gene function, especially in plants that lack an Agrobacterium tumefaciens-mediated transformation system. By establishing this system, the researchers have provided a valuable tool for investigating the molecular mechanisms of mango.
            One of the key findings of this study is the identification of the most beneficial protoplast isolation conditions. The researchers determined that using 0.6 M mannitol and 0.4 M cellulase in 25 ml of enzyme solution, along with a 12-hour digestion time, resulted in the highest yield of leaf protoplasts (approximately 5.4 x 105 cells/10 ml). This optimization of isolation conditions is crucial for obtaining a sufficient number of protoplasts for subsequent experiments.
            Furthermore, the authors successfully transfected the isolated protoplasts using a 40% calcium chloride-based polyethylene glycol (PEG)-4000-CaCl2 method, achieving a remarkable transformation efficiency of 84.38%. The confirmation of successful transformation was demonstrated by the observation of green fluorescent protein (GFP) under ultraviolet light, indicating that the target gene (GFP) was transcribed and translated. This outcome validates the functionality of the established protoplast transformation system.
            The implications of this research are far-reaching. The high-efficiency protoplast isolation and PEG-mediated transformation protocol presented in this study provide a new avenue for gene function research in mango. This system can be effectively utilized to study the molecular mechanisms of mango, enabling investigations into subcellular protein localization, protein-protein interactions, gene function identification, and the revelation of molecular regulatory mechanisms.
            In conclusion, the manuscript "A Protoplast Transformation System to Enable Functional Studies in Mangifera indica L." presents a comprehensive and practical study. The authors have successfully developed an efficient protoplast isolation and transformation system for mango, which holds great potential for advancing gene function research in this fruit tree. The findings of this study contribute significantly to the field and open up new possibilities for investigating the molecular mechanisms underlying the traits and characteristics of mango.
In addition to the existing review, it is worth mentioning a chapter on optimization in the manuscript. The authors refer to the concept of "optimization," but the methodology lacks specific information about the statistical model employed in this study. However, it is important to note that for future experiments, it would be beneficial to design them with optimization in mind, such as employing response surface methodology (RSM).
            Optimization plays a crucial role in experimental design, as it allows researchers to systematically explore and identify the optimal conditions for achieving the desired outcomes. By utilizing RSM, which is a statistical technique used to model and optimize complex processes, researchers can efficiently navigate the experimental parameter space to identify the optimal combination of variables.
            RSM involves the creation of a mathematical model that describes the relationship between the response variable and the experimental factors. This model enables researchers to estimate and predict the response at different factor levels, ultimately aiding in the identification of the optimal conditions for maximum yield, efficiency, or any other desired outcome.
            By incorporating RSM into the experimental design phase, researchers can effectively streamline their efforts and reduce the number of required experiments. This approach not only saves time and resources but also enhances the likelihood of obtaining reliable and significant results.
            Therefore, for future studies aiming to optimize the protoplast isolation and transformation system in mango, it is recommended that the researchers consider employing response surface methodology. By doing so, they can systematically investigate the influence of various factors on the desired outcome and identify the optimal conditions that maximize the efficiency and yield of the protoplast transformation process.

            In conclusion, while the current manuscript lacks specific information regarding the statistical model used for optimization, it is advisable for future researchers to embrace methodologies like response surface methodology to optimize experimental conditions. Implementing optimization techniques can significantly enhance the efficiency and effectiveness of the research, leading to valuable insights into the gene function and molecular mechanisms of mango.

In summary, the reviewed manuscript, "A Protoplast Transformation System to Enable Functional Studies in Mangifera indica L.," presents a valuable contribution to the field of molecular sciences. The development of an efficient protoplast isolation and transformation system for mango provides researchers with a powerful tool for investigating the gene function and molecular mechanisms of this perennial fruit tree.

The manuscript successfully outlines the optimization of protoplast isolation conditions and demonstrates the high transformation efficiency using a calcium chloride-based polyethylene glycol method. The confirmation of successful transformation through the observation of green fluorescent protein further validates the functionality of the established system.
            However, it is important to note that the manuscript would benefit from the addition of information regarding the optimization methodology employed by the authors. Specifically, details about the statistical model used for optimization, such as response surface methodology (RSM), would enhance the scientific rigor and provide a comprehensive understanding of the experimental design.
            Once this revision is implemented, the manuscript will be suitable for publication in the prestigious "International Journal of Molecular Sciences," given its significant contribution to the field and the practical implications for gene function research in Mangifera indica L.
            In conclusion, the manuscript presents a valuable and promising study, offering an efficient protoplast transformation system for mango. With the recommended addition of information regarding the optimization methodology, particularly the use of techniques like response surface methodology, the manuscript will be well-rounded and ready for publication in the esteemed "International Journal of Molecular Sciences."

Responds:

Thank your reviewer for this insightful idea and recommendation of response surface methodology (RSM) as a simple tool to validate and obtain optimal values for the study.

The protoplast transformation technology system is an excellent method for a rapid and effective tool for genetic transient expression and transformation assays studying gene function, particularly in plants that lack an Agrobacterium-mediated plant transformation tumefaciens-mediated trans-formation system. This study developed an efficient protoplast isolation and transformation system, which can help a lot in the gene function expression study of mango. The most beneficial protoplast isolation conditions were 0.00375 mg/ml cellulase R-10 and 0.0045 mg/ml0.4 macerozyme R-10 of the digestion solution and 12 hours of digestion time. The yield leaf protoplasts (~approximately 5.4 x 105 cells/10 ml) were transfected for 30 minutes using 40% calcium chloride-based polyethylene glycol (PEG)-4000-CaCl2, from which 84.38% of the protoplast cells were transformed. Under ultraviolet light the florescence, the GFP in the transformed protoplasts was observed, confirming that the target gene (GFP) is transcribed and translated. The high-efficiency protoplast isolation and PEG-mediated transformation protocol presented herein provided a new way for excellent method for a rapid and effective tool gene function research for research in mango that can be effectively used to study the plant genetic molecular mechanisms of mango.

Our study used different time points for a condition to obtain the optimal values. That is to say, we didn’t maintain each value as constant, but the conditions were such that it was important to consider the study using an ANOVA analysis to compare the means of multiple pairwise comparisons from which the optimal numbers were determined. Line152. Moreover, the suggested RSM is the best fit for our studies and is the same as the method used for these current studies. Moreover, the efficiency and optimal values to validate the observed factors were based on the composition of the enzyme solution (cellulase R-10, macerozyme R-10, and mannitol), digestion time, pH, polyethylene glycol (PEG) concentration, and transformation time. Each factor was analyzed independently to reveal the maximum point. Thus, the response surface methodology (RSM), as stated, is here described in the manuscript as the probable factorial variables, which included digestion and transformation conditions such as mannitol, cellulose, pH, time, and PEG concentrations maximum and minimum values and points to obtain a good digestion solution and time for successful transformation expression. Line 93-122.

The current study is important for molecular mechanisms because the expression at the cell level will give a clear view of how cellular mechanisms evolved, which is crucial to understanding the regulation of photosynthesis, metabolism, and other underlying plant growth mechanisms [28]. The constructed the effective technique of protoplast isolation and transformation protocol. This protocol is for specific gene expression analysis, which can be later use analyze the function of cell developmental genes by observing the growth, analyze the function and development of transformed protoplasts [5]. This protoplast transient expression system worked successfully in investigating subcellular localization and regulating gene expression. Taken together, the development of a mango protoplast transient expression technology provides a versatile experimental platform to enable molecular, cellular, and functional studies of Mangifera indica L.

Inclusive, your suggestion has been considered and has been noted with thanks.

Round 2

Reviewer 1 Report

Remarks to the author

mg/l and replace it with mg/L throughout the document.

Mannitol concentration mg/mL to changes as ug/mL

Space is needed Within lines 106, 217, 218, 129, 226, 245, 247, 250, 255, 256, 299, 308, 311, 372, 409, 412, 413, and 426.

The captions of Figures 5 and 6 should include the scale bar unit. 

Authors should thoroughly examine their entire manuscript for typographical and unit errors. Check the figure's title, symbols, and characters.

Author Response

Dear Sir/Madam thank you for your insightful comments on our work, please find below the responds to your comments.

Comment: mg/l and replace it with mg/L throughout the document.

Responds: The mg/ml has been replace mg/mL throughout the manuscript. 

Comment: Mannitol concentration mg/mL to changes as ug/mL

Responds: All the mannitol concentration the manuscript has been changed from mg/mL to ug/mL

Comment: Space is needed Within lines 106, 217, 218, 129, 226, 245, 247, 250, 255, 256, 299, 308, 311, 372, 409, 412, 413, and 426.

Responds: line spaces has been created in between lines 106, 217, 218, 129, 226, 245, 247, 250, 255, 256, 299, 308, 311, 372, 409, 412, 413, and 426.

Comment: The captions of Figures 5 and 6 should include the scale bar unit. 

Responds:  Figure 5 and 6 in the manuscript  scale bar units of = 100 µm as been added

Comment: Authors should thoroughly examine their entire manuscript for typographical and unit errors. Check the figure's title, symbols, and characters.

Responds: All the grammatical, typographical error, unit errors, figure tittle, symbols, ans reference in the manuscript have been thoroughly examine and improved.

Reviewer 2 Report

Some corrections were done, BUT text require edition by professional edotor as grammar and contents.

The name "mango molecuar mechanism" does not have sense.

Protoplasts represent highy stressed cells without any link with gene expression in planta what is cell type specifis. Please, discuss this.

Authors should ask professional editors to edit paper before re-submission.  All  concentrations were wrong, indeed.

Do you mean for „investigation of molecular mechanism“? Please, explain how you can study photosynthesis in protoplasts? Please, explain which type of the cell represent protoplasts? Molecular mechnasim have sense only in relation with certain cell types /metabolism. However, protoplasts represnt highly stressed cell (cuting, incubation in enzymes, centrifugation) which do not have normal metabolism and „normal“ molecular mechanism.

Line 19- 20: please, corrected numbers: 0,00375 mg/ml??

Line 24: „under fluoresence“?

Line 27: „method for a rapid and effective tool“??

Line 37: „molecular mechanisms of mango.“ ? Do you mean mechanism of gene expression?

Line 64: „dissolves plant cell walls“?  Digest plant cell wall.

Lines 88-100: this part is redudndant here. It looks like M&M. (or discussion, if you compare with your parametrs).

Line 112: „clean by washing with ethanol“ ? Concentration? Duration? Have you washed leaf with water after ethanol? Do not forget that ethanol also make some extent of hydrophilization.

Lines 115-120_ all numbers were wrong. Perhaps, you mean g/ml?

0.002 mg/ml mean 2 mg/l!

Line 128: „digestion suspension“?

Lines 135-137, 141-145: numbers! Please, keep mannitol and Ca in M.

Line 184: How you can observe trypan blue under confocal? it does not require confical, simple DIC is enough!

Line 200: „protocol for isolating and genetically transforming protoplasts“???

Lines 294 – 300: numbers, grammar.

Significant corrections are required-

Author Response

Dear Sir/Madam thank you for your insightful comments on our work, please find below the responds to your comments.

Some corrections were done, BUT text require edition by professional edotor as grammar and contents.

Comment: The name "mango molecuar mechanism" does not have sense.

Responds: The phrase molecular ‘mango molecular mechanism’ stated  in the manuscript means exploring rigorously by genetic and biochemical testing. However it is changed to Tissue culture and gene transformation techniques are limited, which delays the understanding of the studies showed  gene expression genetic and biochemical testing leading to transformation technology of mango. Line 33-35.

Comment: Protoplasts represent highy stressed cells without any link with gene expression in planta what is cell type specifis. Please, discuss this.

Responds:  In an effort to identify potential factors contributing of plant protoplasts in oxidative stress which consist cell wall reconstitution, differentiation, and cell cycle progression [6-8]. In this regard, protoplasts are particularly useful to address essential biological questions regarding stress response, such as protein signaling, ROS production, and plasma membrane dynamics. Particularly in Verticillium dahliae, they used synthetic siRNA targeting the GFP gene in the GFP-transformed phenotype and the Vta2 gene, a regulatory gene essential for growth. Stressed wild-type strain using PEG-mediated transformation to siRNA can enter the protoplasts and inhibit the expression of these genes []

Comment: Authors should ask professional editors to edit paper before re-submission.  All  concentrations were wrong, indeed.

Responds: All the concentrations were calculated in gm/mL and diluted in 25 ml double distilled water.

Comment: Do you mean for „investigation of molecular mechanism“? Please, explain how you can study photosynthesis in protoplasts? Please, explain which type of the cell represent protoplasts? Molecular mechnasim have sense only in relation with certain cell types /metabolism. However, protoplasts represnt highly stressed cell (cuting, incubation in enzymes, centrifugation) which do not have normal metabolism and „normal“ molecular mechanism.

Responds: The phrase investigation of molecular mechanisms" in this statement means that the protoplast can be used for investigating gene expression genetic studies. Which means that a photosynthesis gene can be used here since the protoplast cells and associated organelles  control most photosynthesis activities in plants. More importantly, protoplasts have chloroplasts, and chloroplasts store chlorophyll, and plants use chlorophyll for photosynthesis activities. So putting this together will enable photosynthesis studies. As stated in line 34, the molecular mechanism stated here means the gene expression mechanism.

Comment: Line 19- 20: please, corrected numbers: 0,00375 mg/ml??

Responds: The actual number is 0. 00375 mg/mL. The concentrations were calculated in gm/mL and diluted in 25 ml double distilled water.

Comment:Line 24: „under fluorescence“?

Responds: What we mean is that the fluorescence signal we observed under the microscope confirms that the target gene is transcribed and translated.

Comment: Line 27: „method for a rapid and effective tool“??

Responds: Yes, the protoplast transformation is an effective and rapid tool for alternative plant genetic studies as even described in other studies [6-8].

Comment: Line 37: „molecular mechanisms of mango.“ ? Do you mean mechanism of gene expression?

Responds: The molecular mechanisms of mango means gene expression mechanism. The studies showed  gene expression genetic and biochemical testing leading to transformation technology of mango. Line line 33-35.

Comment: Line 64: „dissolves plant cell walls“?  Digest plant cell wall.

Responds: The phrase means plant cell walls dissolves, breakdown or digest plant cell walls. Line 58.

Comment: Lines 88-100: this part is redudndant here. It looks like M&M. (or discussion, if you compare with your parametrs).

Responds: Line 88-100 was meant to state the actual scope of the work. However due to its redundancy we have sumerize it. Moreover in the discussion, we have compared our finding with other studies to reveal the relevance of the current study. Line 273-382.

Comment: Line 112: „clean by washing with ethanol“ ? Concentration? Duration? Have you washed leaf with water after ethanol? Do not forget that ethanol also make some extent of hydrophilization.

Responds: We didn’t use 95% ethanol and washed with running water. We agree the ethanol make some extent of hydrophilization. Before use the cell strainer was kept in 95% ethanol and washed in water before use to prevent infection of the cells. Line 119-121.

Comment: Lines 115-120_ all numbers were wrong. Perhaps, you mean g/ml?

Responds: Thank you sir, because most of the reagent used was expensive, from which we can afford 1g per bottle to prepare four solution. We weighed small portion in grams and converted to mg to help present our findings very well. This is to say, line 115-120 are measure in milligram/milliliters. Moreover, to omitted most of the zero’s the mannitol is converted from mg/ml to ug/ml to make simple.

Comment: 0.002 mg/ml mean 2 mg/l!

Responds: Thank you, sir, because most of the reagent used was expensive, and we can afford 1g per bottle to prepare four solutions. We weighed small amounts, such as 0.2 g, converted mg to portions, and diluted them in 25 ml of ddH2O to help present our findings very well. This is to say, lines 115–120 are measured in milligram/milliliters. Moreover, to omit most of the zeros, the mannitol is converted from mg/ml to ug/ml to make it simple.

Comment: Line 128: „digestion suspension“?

Responds: Line 128, the digested suspension mean that the digested protoplast cell mixed in digestion solution.

Comment: Lines 135-137, 141-145: numbers! Please, keep mannitol and Ca in M.

Responds: Thank you, for your insightful comment on the number, however we have kept the mannitol in ug/ml and Ca mg/ml because the amount used is small and it will be much clearer. Thank you.

Comment: Line 184: How you can observe trypan blue under confocal? it does not require confical, simple DIC is enough!

Responds: The trypan blue stained cell was observed under normal light on a microscope. Line 160. Thank you

Comment: Line 200: „protocol for isolating and genetically transforming protoplasts“???

Responds: In this study, we developed the protocol for the isolation of protoplast cells from leaf tissue, which can be used for gene expression transformation studies using PEG as the transformation medium Line 101-129. In this, the possible factors to obtain the highest protoplast cells were considered, from which a gene was used to test the cell potency to ensure a successful gene expression technique using a GPF tag. Line 131-151 

Comment: Lines 294 – 300: numbers, grammar.

Responds: Line 294-300 have been revised to Under florescence light, the green fluorescent protein (GFP) signal can be clearly observed in the protoplast. The GFP signal appeared after 20 minutes of transfection using 40% PEG-4000-CaCl2 (Figure 5C), and a stronger GFP signal was observed after 30 minutes of transfection (Figure 5D). The GFP signal observed under the florescence light of the protoplast showed that the successfully transformed protoplast expressed effectively.

Round 3

Reviewer 2 Report

Line 40: citation 6 does not relate with message you mention. Plesae, check all citation and fit all of them.

Concentration: mannitol: 0.2 M = 36 g/l, but you wrote 0.2mkg/ml. Do not confuse molar concentration and mg/l!

Enzyme: the working concentrations of enzymes (macerozyme, cellulase) are 0.4- 1% what mean 4-10 mg/ml. So, how can you increase enzyme activity 1000 times? Please, either explain this phenomena, or write CORRECT concentrations. The same for all other ions.

Line 145: KCl2 is NOT exist!

Figure 5; where is a control? Which wavelength for exitation and emissión?   

Responds:  In an effort to identify potential factors contributing of plant protoplasts in oxidative stress which consist cell wall reconstitution, differentiation, and cell cycle progression [6-8]. In this regard, protoplasts are particularly useful to address essential biological questions regarding stress response, such as protein signaling, ROS production, and plasma membrane dynamics. Particularly in Verticillium dahliae, they used synthetic siRNA targeting the GFP gene in the GFP-transformed phenotype and the Vta2 gene, a regulatory gene essential for growth. Stressed wild-type strain using PEG-mediated transformation to siRNA can enter the protoplasts and inhibit the expression of these genes.”

This is correct, BUT in your case you never use cell wall reconstruction, cell cycle etc. Without these steps, it is only “empty ballon” with very limited gene expression pathways. Moreover, in your case protoplasts lack all significant ions like Cu, Zn etc what is cofactor for many response pathway. If you will use full médium, you can make this statement.

“gene expression genetic studies” – this is NON-sense. Gene expression studies is enough.

Responds: The phrase investigation of molecular mechanisms" in this statement means that the protoplast can be used for investigating gene expression genetic studies. Which means that a photosynthesis gene can be used here since the protoplast cells and associated organelles  control most photosynthesis activities in plants. More importantly, protoplasts have chloroplasts, and chloroplasts store chlorophyll, and plants use chlorophyll for photosynthesis activities. So putting this together will enable photosynthesis studies. As stated in line 34, the molecular mechanism stated here means the gene expression mechanism.“

Photosyntehsis require many ions like Cu, Zn etc and high light intensity. Which light intensity have you used? High light bis lethal for protoplasts.

“Under florescence light, the green fluorescent protein (GFP) signal can be clearly observed in the protoplast.”  - please, provide wavelength for Ex and Em!

20 min is too short period, please, provide a control image: prptpalsts transformed with empty vector and took signal with the same parameters as GFP-treated cells.

major corrections

Author Response

Dear Sir/Madam thank you for your insightful comments on our work, please find attached below the responds to your comments.

Thank you.

Round 4

Reviewer 2 Report

The authors need to provide the negative control for GFP expression (20 min is too short to induce full expression).

Also, authirs need to repeat experiments by using the concentration of the chemicals they mention like "0.00375 mg/mL cellulase, 0.003 mg/ml Mannitol as osmotic etc.

In all other species the concentration of Mannitol must be 72 g/l (0,4 M) and enzyme cellulase concentration 4 mg/ml or so.

The question still open: does Mango require such low ions/osmotic or it is wrong calculation made by authors.

corrections are required

Author Response

Dear Reviewer,

Thank your your insighful comments. Below is the responnse

Comment: The authors need to provide the negative control for GFP expression (20 min is too short to induce full expression).

Responds: Thanks for your patient review. According to your suggestion, we have added the negative control. We have included the negative control in statement. This is because there was no signal of GFP florescence to show. The full expression of the GFP signal was not 20 minutes, but the transfection period was conducted on six different time period (5, 10, 20, 30, 40, and 60) min and after which it was left in the dark for 16 to 20 hours, before the subcellular localization of the GFP signal in the transformed protoplast fluorescence images was photographed.

Comment: Also, authors need to repeat experiments by using the concentration of the chemicals they mention like "0.00375 mg/mL cellulase, 0.003 mg/ml Mannitol as osmotic etc.

Responds: Thanks for your concern.All the osmotic substance and reagents used was measured in grams and diluted in 25 ml water.After that we presented out results in mg/ml due to the quantity of the water used. We hope that these revisions text will be satisfactory.

Comment: In all other species the concentration of Mannitol must be 72 g/l (0,4 M) and enzyme cellulase concentration 4 mg/ml or so.

Responds: Thank you for comment, we scaled down all the digestion solution component to small portions. All the mannitol concentration has been changed back (M). Moreover, we believe your suggestion is insightful however the company we buy from don't produce these substance in large volume so we prepare according to what we buy. We used two osmotic digestion substances since we used smaller portions. This is because the cellulase R-10 and  macerozyme R-10 are expensive from which we are able to get  1g and 10 g from the company respectively. Due to this we scaled the all to meet the cellulase R-10 and  macerozyme R-10. Though believe other studies use larger portion and presented their results in results concentration in M, g/l and mM. We believe that our result can also be presented in these formats but we choose these due to their small quantities used.We hope that these revisions text will be satisfactory.

Comment: The question still open: does Mango require such low ions/osmotic or it is wrong calculation made by authors.

Responds: Thanks for your patient review. The important work in this study is the study of the effects of these osmotic substances on a larger quantity of protoplasts from mango leaf tissues. As presented in the materials and methods, the osmotic substances in the digestion solution are not low because, the amount of water used to dilute the solutes was small (25 ml) and the portion was concentrated enough to digest the cell wall, where we were able to obtain a higher protoplast cell at 7.1 x 105. This means that a larger amount in liters (L) will require a higher amount (g) of substance, and vice versa.Similar component was used in ‘Arabidopsis mesophyll protoplasts: a versatile cell system for transient gene expression analysis. Nature protocols 2007, 2”[6].

Thank you